# Pulmonary Hypertension in Interstitial Lung Disease: Updates in Disease, Diagnosis, and Therapeutics

**DOI:** 10.3390/cells12192394

**Published:** 2023-10-01

**Authors:** Zachary A. Haynes, Abhimanyu Chandel, Christopher S. King

**Affiliations:** 1Department of Pulmonary and Critical Care, Walter Reed National Military Medical Center, Bethesda, MD 20889, USA; 2Advanced Lung Disease and Transplant Program, Inova Heart and Vascular Institute, Inova Fairfax Hospital, Fairfax, VA 22031, USA; christopher.king@inova.org

**Keywords:** pulmonary hypertension, interstitial lung disease

## Abstract

Pulmonary hypertension is a debilitating condition that frequently develops in the setting of interstitial lung disease, likely related to chronic alveolar hypoxemia and pulmonary vascular remodeling. This disease process is likely to be identified more frequently by providers given recent advancements in definitions and diagnostic modalities, and provides practitioners with emerging opportunities to improve patient outcomes and quality of life. Despite years of data suggesting against the efficacy of pulmonary vasodilator therapy in patients with pulmonary hypertension due to interstitial lung disease, new data have emerged identifying promising advancements in therapeutics. The authors present to you a comprehensive review of pulmonary hypertension in interstitial lung disease, reviewing our current understanding of pathophysiology, updates in diagnostic approaches, and highlights of recent clinical trials which provide an effective approach for medical management.

## 1. Introduction

Pulmonary hypertension (PH) is an often progressive and debilitating condition comprising a diverse collection of disease processes, characteristically due to elevated pulmonary artery pressures from either precapillary, postcapillary, or mixed etiologies. This framework allows for the stratification of a patient’s underlying etiology of PH into one of five groups based on pathophysiology utilizing the World Health Organization (WHO) classification system. WHO Groups 1, 3, and 4 represent precapillary causes, Group 2 represents postcapillary causes, and Group 5 represents diseases with multifactorial etiologies [1]. Specifically, patients with Group 3 PH demonstrate elevated pulmonary artery pressures due to chronic lung disease or chronic hypoxia, and, classically, this group demonstrates the greatest morbidity and mortality [2,3]. PH due to interstitial lung disease (ILD; PH-ILD) represents a subset of Group 3 and presents clinicians with a unique set of diagnostic and management challenges.

The recently gathered 6th World Symposium of Pulmonary Hypertension defined PH utilizing a mean pulmonary artery pressure (mPAP) of greater than 20 mmHg, which was reduced from the prior cut-off threshold of 25 mmHg to improve diagnostic sensitivity [4,5]. More specifically, Group 3 PH can be identified when a patient presents with isolated precapillary PH, defined as mPAP > 20 mmHg, pulmonary vascular resistance (PVR) > 3 Wood units (WU), and pulmonary capillary wedge pressure <15 mmHg in the presence of underlying lung disease or hypoxia. The European Society of Cardiology (ESC) and the European Respiratory Society (ERS) subsequently have proposed an alternative definition for precapillary PH, recommending a PVR > 2 WU as an alternative diagnostic cut-off [6]. These increasingly inclusive definitions will undoubtedly increase the identification of PH-ILD and allow for treatment to be employed at an earlier stage, hopefully mitigating the downstream consequences of disease.

The development of PH-ILD can be quite common in patients with ILD. Prevalence is thought be variable among different subsets of ILD and increases with disease severity, with the bulk of our understanding deriving from patients with idiopathic pulmonary fibrosis (IPF). Patients with IPF have been cited as having anywhere from 8% to 86% prevalence of PH-ILD, with this wide range likely being related to prior variability in the definition of PH and the methods used for diagnosis. General consensus among PH experts places overall prevalence somewhere between 30 and 50% [7]. At diagnosis, 8–15% of patients with IPF may demonstrate PH-ILD, whereas up to 50% of advanced cases and more than 60% of end-stage cases will demonstrate PH-ILD [8,9]. Several smaller studies have described the prevalence of PH-ILD in other ILD phenotypes, notably nonspecific interstitial pneumonia and chronic hypersensitivity pneumonitis, with 31.4% and 44% prevalence reported, respectively [10,11]. Data beyond this on other fibrotic ILD are limited and need further investigation.

Among patients with ILD, the identification of PH typically is a finding of grave consequence. PH-ILD portends increased morbidity with decreased functional capacity, more frequent hospitalizations, higher supplemental oxygen requirements, and reduced quality of life when compared to patients with ILD without PH [9,12]. This negative relationship additionally holds true for mortality, where patients with PH-ILD may experience as much as a five-fold increase [13]. Though survival is variable among various subgroups of ILD, once patients develop PH-ILD, survival appears to be similar and related to the degree of elevation in pulmonary artery pressures [11,14]. This suggests that PH tends to drive these patients’ clinical trajectory, highlighting the importance of effective screening, and ultimately management, for improving outcomes. Given this information, we subsequently describe the causative pathophysiology of the disease, provide a diagnostic framework, and discuss the recent advances and discussions surrounding therapy for PH-ILD.

## 2. Pathophysiology

The development of PH-ILD is complex and not fully understood, though chronic alveolar hypoxia is likely a key mediator. Unlike systemic vascular structures which dilate in response to decreased blood oxygen tension, pulmonary vascular tone increases in response to low oxygen tension in a process called hypoxic pulmonary vasoconstriction (HPVC). Hypercapnia or acidemia can additionally augment the response of the pulmonary vascular bed to hypoxemia. Healthy pulmonary tissue performs HPVC in an effort to maintain systemic arterial oxygenation by diverting blood flow from regions of relative hypoxia and proportionately towards regions of normal oxygenation, improving ventilation/perfusion matching. This adaptive mechanism is intended to correct oxygen saturation, so patients may initially demonstrate near-normal oxygen saturation despite large regions of fibrosis and alveolar hypoxia if they are appropriately compensated [12].

The effects of HPVC can typically be reversed with the implementation of supplemental oxygen, though repetitive exposure to hypoxia can lead to vascular remodeling of peripheral vessels and increased smooth muscle in the pulmonary vascular tissues. Vascular remodeling occurs through various mechanisms in the pulmonary vasculature, notably with muscularization of the media and thickening of the intima of small-to-medium pre-capillary pulmonary arterioles and arteries [15]. Though these adaptations may initially improve oxygenation, over time they can become maladaptive and result in a diminished HPVC response to hypoxia, ultimately leading to worse systemic oxygenation due to ventilation/perfusion mismatch (Figure 1) [16]. Vascular remodeling can also lead to a phenotype reminiscent of pulmonary arterial hypertension, with PH out of proportion to the degree of lung disease identified [17]. The effects of HPVC on PH are likely further exacerbated by parenchymal destruction in the setting of fibrotic lung disease, reducing the quantity of vessels available for gas exchange and limiting the ability of the remaining vessels to dilate in response to increased cardiac output during activity [18]. All of these effects ultimately increase PVR and worsen PH.

HPVC and damage to the pulmonary vascular bed are unlikely to solely account for the explanation of PH-ILD development (Figure 1). The lack of correlation between the degree of restriction on spirometry and severity of PH supports this idea and raises questions surrounding what factors may be promoting disease [12]. Disrupted angiogenesis can be seen in regions of fibrosis and levels of profibrotic or proinflammatory mediators, including tumor necrosis factor, platelet-derived growth factor, fibroblast growth factor, profibrogenic leukotrienes, and transforming growth factor B may be elevated [19,20]. It is noteworthy that vascular remodeling can also be seen in histopathology samples of regions of lung which are left unaffected by fibrosis, supporting a more systemic driver to this pathology [12].

In response to increasing PVR, patients with PH-ILD commonly demonstrate progressive right ventricular (RV) dysfunction. Relative to the left ventricle, the thin-walled RV is highly sensitive to hemodynamic changes and maintaining function is exquisitely important to support systemic cardiac output due to ventricular interdependence. With the progression of PH-ILD, RV afterload significantly increases due to increasing PVR, promoting progressive RV dysfunction. Chronically increased afterload leads to RV hypertrophy through neurohormonal activation with increased adrenergic tone [21]. This is initially adaptive, preserving stroke volume and increasing contractility, though over time beta adrenergic receptor density declines, adrenergic effectors are depleted, and tissues become less responsive to beta adrenergic agonists, forcing the RV to dilate to maintain stroke volume [22]. These processes ultimately lead to increasing RV wall stress, promoting RV ischemia and further declines in function. Continual stress, ischemia, and neurohormonal activation promote fibroblast collagen proliferation, resulting in progressive fibrosis, impaired diastolic relaxation, and diminished contractility [21]. Volume retention, worsening hypoxemia, and low cardiac output can result from this remodeling, placing patients with PH-ILD in the unfortunate scenario of right ventricular failure.

Although it is clear that advanced PH-ILD leads to RV dysfunction and this drives worsening outcomes, why relatively modest increases in pulmonary artery pressures (PAPs) that do not manifest in overt RV dysfunction are associated with such poor outcomes is less clear. It is possible that these patients rapidly progress to have more severe PH-ILD and studies fail to capture this progression over time. Another possibility is that these patients have exercise-induced PH that causes RV strain with activity. A final possibility is that these patients suffer from a relatively low cardiac output state that accounts for an elevated PVR despite relatively low mean PAPs. These conditions could limit oxygen delivery to tissues with exertion. Further study of patients with mild PH-ILD to determine the response to exercise and natural history is essential in answering these questions. Furthermore, whether treatment with pulmonary vasodilators will halt the progression of mild PH-ILD and improve outcomes requires formal evaluation in clinical trials.

## 3. Diagnosing a Case of PH-ILD

Unfortunately, there is limited information on which patients should be screened for PH-ILD and a paucity of consensus guidelines exist to direct a clinician’s diagnostic workup. The symptoms of PH-ILD have significant overlap with those of fibrotic lung disease and only a portion of patients with ILD may go on to develop PH-ILD, further complicating the diagnostic process [17]. Additionally, even mild elevations of pulmonary pressures are clinically impactful and associated with adverse outcomes; however, these may not manifest in overt changes on physical exam or even transthoracic echocardiogram [23]. Physical exam is typically unrevealing until patients develop more advanced PH and a clinician’s ability to diagnose PH from signs and symptoms alone has been demonstrated to be inadequate [24]. Despite this, a recent Delphi study came to a consensus on symptoms or historical features which warrant further screening for PH when present in patients with ILD [25]. This can be used to select patients for screening in conjunction with other factors previously demonstrated to suggest the presence of PH-ILD (Table 1) [17].

Given the challenges of identifying PH-ILD from history or exam, clinicians must maintain a high index of suspicion. Careful review of non-invasive parameters may be suggestive of the diagnosis of PH-ILD. Chest CT can be a useful tool for both monitoring ILD progression and identifying PH-ILD. One notable CT finding is a pulmonary artery to aorta diameter ratio of >0.9, which has been suggested to predict a mPAP of >20 mmHg and reduced survival [26]. Identification of an RV diameter greater than the LV diameter has also been associated with elevated PVR in patients with ILD [27]. In regard to biomarkers of disease, brain natriuretic peptide (BNP) and N-terminal (NT) proBNP appear to have diagnostic utility. An NT-proBNP level of <95 ng/L has previously been suggested as a rule-out criteria for PH-ILD with a negative predictive value of 99% [28]. Alternatively, an elevated BNP or NT-proBNP value should trigger suspicion for underlying PH-ILD, but is nonspecific [25]. It should be noted that obesity can lead to pseudo-normalization of BNP and pro-BNP values and can limit the utility of these assays as a screening tool [29].

Pulmonary function testing (PFT) has limited utility in the identification of PH-ILD, though diffusion capacity of carbon monoxide (DLCO) appears to have both diagnostic and prognostic utility. Severely reduced (<30% of predicted) DLCO has been shown to increase the likelihood of PH-ILD twofold [30]. Reduced DLCO is also the only PFT variable shown to be a marker of mortality in Group 3 PH, with each 10% reduction from predicted DLCO being associated with a 31% increased risk of mortality [31]. Basic spirometry has not been shown to be useful in identifying underlying PH-ILD [12]. The six minute walk test can also be used as a rough screening tool for PH-ILD, with a significantly reduced walking distance or marked desaturations being associated with development of PH-ILD [32]. Additionally, heart rate recovery (the reduction in heart rate from the maximum heart rate after 1 min of rest) of less than 13 beats has been shown to be predictive of PH-ILD in a cohort of patients with IPF [33].

In general, we recommend the regular use of transthoracic echocardiography (TTE) to screen for PH-ILD in patients thought to be at risk based off of history, exam, and diagnostic findings, in concert with a careful review of PFTs, cross sectional imaging, and exercise data. We also recommend to repeat TTE annually in patients with an established diagnosis. Screening should be serial in nature given the progressive nature of ILDs, as the incidence of PH-ILD may as much as double over five years of follow up [8]. Historically, PH has been screened via TTE by calculating the RV systolic pressure (RVSP) indirectly through measurement of the tricuspid regurgitant jet velocity (TRV), with an RVSP > 35 mmHg being elevated. Recent updates in the ESC/ERS Guidelines for the Diagnosis and Treatment of Pulmonary Hypertension favor following the TRV as opposed to the calculated RVSP, with a TRV > 2.8 m/s being suggestive of PH [6]. These methods can unfortunately be inaccurate in patients with ILD due to difficulties visualizing the regurgitant jet [34]. Advances in echocardiography have also allowed for the measurement of other right ventricular parameters, including the tricuspid annular plan excursion, fractional area of change, and RV outflow tract diameter, which appear to provide a more accurate hemodynamic assessment for PH-ILD and can additionally be used prognostically [35,36]. Notably, a fractional area of change derived from three-dimensional echocardiography below 28% has been shown to place patients with PH-ILD at increased risk for hospitalization and death [35]. As mentioned previously, given the relatively modest elevations required to secure a diagnosis of PH-ILD and the technical difficulties in visualizing the RV in ILD, TTE findings may be subtle or absent. In fact, Keir and colleagues found that 40% of patients deemed low risk for PH by the ESC/ERS TTE screening recommendations actually had PH-ILD when RHC was performed [37].

Given the inadequacies of noninvasive diagnostic strategies, the gold standard for confirming a diagnosis of PH-ILD remains right heart catheterization (RHC) [6]. We recommend any patient with TTE findings of PH-ILD should be referred for RHC (Figure 1). In addition, patients with suggestive findings on ancillary testing (Table 1) without an alternative explanation should be referred, even in the face of a “low risk” TTE. RHC results consistent with PH-ILD are the finding of isolated precapillary PH (mPAP ≥ 20 mmHg, PVR > 2 WU, PCWP ≤ 15 mmHg) with evidence of ILD on cross-sectional imaging [4]. Clinicians must remember that an RHC is a single snapshot in time and may misclassify patients as precapillary PH when viewed in isolation. Left heart dysfunction, particularly diastolic dysfunction, may still be present despite a normal PCWP or LV end diastolic pressure and may complicate therapy [38]. The echocardiogram should be carefully reviewed for evidence of LV systolic or diastolic dysfunction. If suspected, a provocative maneuver to “unmask” diastolic dysfunction such as exercise or fluid challenge should be performed during RHC.

## 4. Managing a Case of PH-ILD

Once PH is confirmed in a patient with ILD using RHC, it is imperative to rule out other possible contributors to PH and comorbid conditions prior to initiation of therapy. Serologic testing and historical assessment for disorders which cause pulmonary arterial hypertension (Group 1) should be undertaken if not already carried out, including HIV, thyroid and liver disease, stimulant medications, and autoimmune diseases [6]. Imaging should also be obtained to exclude involvement from chronic thromboembolic pulmonary hypertension (Group 4) [6]. Careful physical examination and review of serologic testing for connective tissue diseases (CTD) is of particular importance, as PH in the setting of CTD-ILD presents a unique challenge. Clinicians must review the hemodynamic data from RHC as well as imaging studies and PFT data, in an effort to determine if the patient has WHO Group 3 PH as opposed to WHO Group 1 PH in the setting of ILD. Unfortunately, there is no universally agreed upon criteria for how to draw this distinction. In general, more severe elevation of PVR and less severe restriction as assessed by CT imaging and PFTs favors a diagnosis of Group 1 PH. These patients are often treated with multiple pulmonary vasodilators [6,9].

PH-ILD presents a potentially unique physiologic challenge with the introduction of pulmonary vasodilators, different from other PH physiologies. Use of these agents poses a theoretical risk of worsening hypoxemia through uncoupling of the ventilation/perfusion (V/Q) ratio in areas with significant fibrosis [39]. Concern for this process originated from a small trial comparing intravenous epoprostenol and sildenafil for management of PH-ILD, where a worsened shunt fraction and hypoxemia were noted in the epoprostenol arm [40]. The data from subsequent trials have been largely unsupportive of this claim, though anecdotally, concern for the physiologic derangement remains [39]. In addition to V/Q derangements, the development of pulmonary edema with pulmonary vasodilators has been suggested as a potential complication of therapy. This is thought to be due to vasodilation of vasculature affected by pulmonary veno-occlusive lesions, which can frequently be seen in patients with ILD, though reports of this outcome are notably rare [41,42].

One potential explanation for the varied responses to therapy described may be the presence of two distinct phenotypes of PH-ILD. In some cases, PH-ILD is likely representative of an adaptive process. The PH in these patients reflects the severity of the underlying fibrotic disease and its resulting alveolar hypoxemia, and PH develops through an attempt to preferentially shunt blood flow to better ventilated regions [17]. These adaptive patients tend to have more mild PH and may be less responsive to pulmonary vasodilators. In other cases, PH-ILD may be a maladaptive process with the degree of PH being disproportionate to the severity of ILD identified, where PH develops from discrete physiologic derangements [17]. These out-of-proportion patients may be more likely to respond favorably to pulmonary vasodilator therapy. Ultimately, the decision to trial therapy should be individualized to each patient based on their clinical characteristics and symptom severity, though pharmacologic options with proven benefit are limited.

### 4.1. A History of Negative Trials

There was initial interest in employing endothelin receptor antagonists (ERAs) for management of PH-ILD. These trials unfortunately resulted in disappointing outcomes. One study on the ERA, bosentan, demonstrated no improvement in pulmonary hemodynamics, respiratory symptoms, or functional status after 16 weeks of therapy [43]. Another trial, Ambrisentan in Subjects With Pulmonary Hypertension Associated With Idiopathic Pulmonary Fibrosis (ARTEMIS-PH), examining ambrisentan in patients with PH due to IPF, was terminated early when its sister study within patients with IPF without PH demonstrated an increased risk of disease progression and hospitalization for respiratory decompensation [44,45]. Additionally, several other recent studies on ERAs in patients with IPF without PH failed to demonstrate superiority of ERAs to placebo for mortality, delayed disease progression, or improved six-minute walk distance (6MWD) [46,47]. Because of these findings, ERAs are typically avoided in the management of PH-ILD.

Commonly used to manage Group 4 PH, the guanylate cyclase stimulator riociguat has also garnered interest in PH-ILD. This was initially spurred by a 2013 open-label, uncontrolled pilot trial of 22 patients with PH-ILD which showed improvements in cardiac output and PVR after 12 weeks of therapy [48]. A larger randomized controlled trial, Riociguat for Idiopathic Interstitial Pneumonia-associated Pulmonary Hypertension (RISE-IIP), followed and ultimately demonstrated an increased risk of mortality and adverse events among patients treated with riociguat when compared to a placebo [49]. Riociguat appears to have an unfavorable risk–benefit profile and its use is recommended against in PH-ILD.

No prospective trial to date has demonstrated positive outcomes from the administration of phosphodiesterase-5 (PDE-5) inhibitors for the management of PH-ILD, though several trials and related registries suggest a potential benefit. Within the Sildenafil Trial of Exercise Performance in Idiopathic Pulmonary Fibrosis (STEP-IPF), sildenafil failed to significantly improve 6MWD, though it was found to improve numerous secondary outcomes, including oxygen saturation and DLCO [50]. Despite these findings, direct application of its results to patients with PH-ILD is difficult, as PH was not confirmed with RHC or TTE in this study and the presence of PH was simply enriched for through the use of DLCO cutoffs for enrollment. Alternatively, the COMPERA registry demonstrated improvements in 6MWD and functional class in patients with PH-ILD, 88% of whom received PDE-5 inhibitor therapy [14]. The subgroup analysis of patients with improved 6MWD from this registry also showed improved survival. Sildenafil has also demonstrated maintained V/Q matching in patients after administration, suggesting that it may preferentially dilate pulmonary vasculature in well-ventilated regions [40]. Two recent studies compared combination of sildenafil with the anti-fibrotic medications pirfenidone and nintedanib versus anti-fibrotic therapy alone. Both studies failed to demonstrate a conclusive improvement in clinical outcomes from the use of sildenafil, although the INSTAGE study of nintedanib/sildenafil noted stabilization of BNP values in those treated with sildenafil [51,52]. The authors of a recent meta-analysis including four studies of sildenafil in IPF suggest a potential mortality benefit to the use of sildenafil in IPF; however, this conclusion should be viewed cautiously, as the data failed to reach statistical significance [53]. Cumulatively, this data suggests that sildenafil is safe and may provide a potential benefit in the management of PH-ILD, though further confirmatory data from prospective trials are needed.

Investigation of all of these agents has been fraught with challenges including wide variability in how PH is defined, small sample sizes, and the abstraction of data from patients with ILD without PH. Given these issues, recommendations for vasodilator therapy for PH-ILD have been limited and without strong evidence until recently.

### 4.2. A Novel Therapeutic

After years of trials without reassuring data, the Inhaled Treprostinil in Pulmonary Hypertension Due to ILD (INCREASE) trial brought forward a novel therapeutic with proven benefit for PH-ILD [54]. The INCREASE trial compared four times daily inhaled treprostinil, a stable analogue of prostacyclin which promotes the local vasodilation of pulmonary vascular beds, to placebo in patients with PH-ILD and found that patients treated with treprostinil demonstrated a significantly increased 6MWD after 16 weeks of therapy [54]. Participants treated with inhaled treprostinil saw a mean increase in 6MWD of 21.08 ± 5.12 m, whereas patients treated with placebo experienced a decrease of 10.04 ± 5.12 m [54]. Additionally, those treated with the study drug saw significant reductions in NT-proBNP levels from baseline and less overall clinical worsening when compared to placebo, further demonstrating benefit [54]. In addition to these promising results, adverse events were generally comparable between the treatment and placebo groups, with only throat irritation and oropharyngeal pain being significantly more common among those treated with inhaled treprostinil [54]. In contrast to other PH therapies, inhaled treprostinil serves as the first agent with demonstrated objective benefit in the treatment of PH-ILD with minimal increased risk to placebo. Given the inhaled nature of this medication, it mitigates the theoretical concern of worsening V/Q matching since the drug would not be significantly delivered to poorly ventilated lung regions. Beyond these initial findings, a subsequent post hoc analysis of INCREASE found improvements in forced vital capacity among subjects treated with treprostinil, which was most pronounced among patients with IPF [55].

### 4.3. An Approach to Management

Studies have noted that even modest elevations in mPAP (>17 mmHg) can lead to increases in mortality by as much as 45.5% [23]. Given the grave prognosis associated with PH-ILD, accurate diagnosis and appropriate management is of paramount importance. The authors believe all patients with evidence of PH-ILD meeting the hemodynamic definition utilized in the INCREASE trial (mPAP > 25 mmHg, PCWP ≤ 15 mmHg, PVR > 3 WU) should be considered for pharmacologic therapy (Figure 2) [6,25]. As the only agent with randomized controlled trial data supporting its safety and efficacy, inhaled treprostinil is the recommended first line therapy in patients with PH-ILD [6,54]. Patients should be titrated to the maximum tolerated dose of treprostinil (up to 64 mcg inhaled four times per day in a dry powder inhaler) due to the greatest clinical benefit being achieved with increased doses of the medication [54]. Notable adverse effects which may limit higher doses include hypotension, bronchospasm, and increased bleeding due to impaired platelet aggregation [54].

Some experts may choose to reserve treatment for patients with severe disease (PVR > 4 WU), as subgroup analysis of the INCREASE trial showed that inhaled treprostinil only demonstrated significant improvement in 6MWD with these patients [54]. As this was not a planned subgroup analysis of the study, the authors would recommend cautious interpretation of these data. A careful consideration of the risks and benefits of inhaled treprostinil should also be given to patients with concurrent PH-ILD and left ventricular diastolic dysfunction before initiating therapy due to this population being excluded from the INCREASE trial. Initiation of pulmonary vasodilator therapy in these individuals can be inherently challenging, with their use potentially worsening left ventricular dysfunction due to increased left ventricular preload. Despite this concern, providers may feel inclined to consider therapy in this group of patients due to limited alternate treatment options. The exact characteristics of patients with favorable risk profiles remains unknown, though patients with mild diastolic dysfunction could conceivably be suited for therapy. Currently, therapeutic decisions must be individualized to each patients’ tolerable risk of decompensation and conceivable benefit from therapy in light of their PH-ILD severity. Close clinical monitoring will be necessary if therapy is offered. Hopefully, further study will shed more light as to the relative risks and benefits of inhaled treprostinil in PH-ILD with various hemodynamic profiles.

Though monotherapy with inhaled treprostinil may be adequate to manage some individuals, patients with more advanced disease with severely increased PVR may require additional pharmacotherapy. Other patients may prove to be intolerant to inhaled treprostinil due to side effects, so alternative therapies may be considered. Recommendations for second line and adjunctive therapy are limited. The use of sildenafil appears to have the potential for benefit and safe use in some patients based upon the results of the COMPERA registry and subgroup analysis from the STEP-IPF trial [14,50]. The authors believe the cautious introduction of sildenafil in PH-ILD is reasonable to try in patients who are intolerant and unable to receive inhaled treprostinil, or as add-on therapy to prostanoids in severe PH-ILD (Figure 2). Further prospective trials on this strategy are needed to better understand the utility of sildenafil and other PDE-5 inhibitors in the management of severe PH-ILD. Additionally, some patients may present with very severe PH-ILD (CI < 2) with precapillary hemodynamics, and our current practice is to treat them, particularly those with mild ILD or CTD-ILD, with parenteral prostanoid therapy and PDE-5 inhibitors. This is often employed as a strategy to bridge them to lung transplantation. This should only be performed at centers with experience in PH, and patients should be carefully monitored for signs of worsening hypoxemia.

Beyond pulmonary vasodilators, management of PH-ILD continues to be anchored in the treatment of the patient’s underlying lung disease and management of contributors to hypoxemia [6]. Patients with hypoxemia at rest, defined as a PaO_2_ < 60 mmHg or SpO_2_ < 92%, should be considered for continuous supplemental oxygen to reduce hypoxic vasoconstriction and improve PVR [56,57]. Additionally, patients should be screened for sleep-related oxygen desaturations and sleep disordered breathing, and be provided supplemental oxygen or ventilatory support as indicated [58,59]. A structured exercise program or pulmonary rehabilitation are often recommended for patients with PH-ILD to mitigate deconditioning and improve patient’s functional status, with significant improvements noted in 6MWD [58,60]. Vaccination for respiratory illnesses, namely pneumococcal pneumonia, COVID-19, and influenza, are also recommended [58,61]. Additionally, if patients present with a fibrotic ILD, antifibrotic therapy with pirfenidone or nintedanib should be offered to slow disease progression and mitigate the impacts of worsening disease [62,63].

In general, all patients with PH-ILD should be considered for early referral to centers of excellence in the management of pulmonary hypertension and advanced lung diseases [6]. Several factors which suggest a need for urgent referral include severe disease (PVR > 5 WU), in addition to decompensated heart failure from severe RV dysfunction. These expert centers are equipped to provide patients access to advanced diagnosis and management strategies, enrollment in clinical trials, and ultimately lung transplant if clinically indicated. The presence of PH in a patient with ILD qualifies them for transplant listing and transplant may ultimately be considered in patients whose disease progresses despite therapy [64].

## 5. Conclusions

PH-ILD is a debilitating disease with significant associated morbidity and mortality. Identifying cases of PH-ILD can be challenging given the overlap of symptoms with their underlying ILD and limitations of current diagnostic strategies. Given these challenges, providers should have a high index of suspicion for PH in patients with ILD and a low threshold to screen these individuals. This is especially important in light of the recent developments in pharmacology for PH-ILD, as inhaled treprostinil appears to be the first safe and effective pharmacologic intervention that can modify disease severity.

## Figures and Tables

**Figure 1 cells-12-02394-f001:**
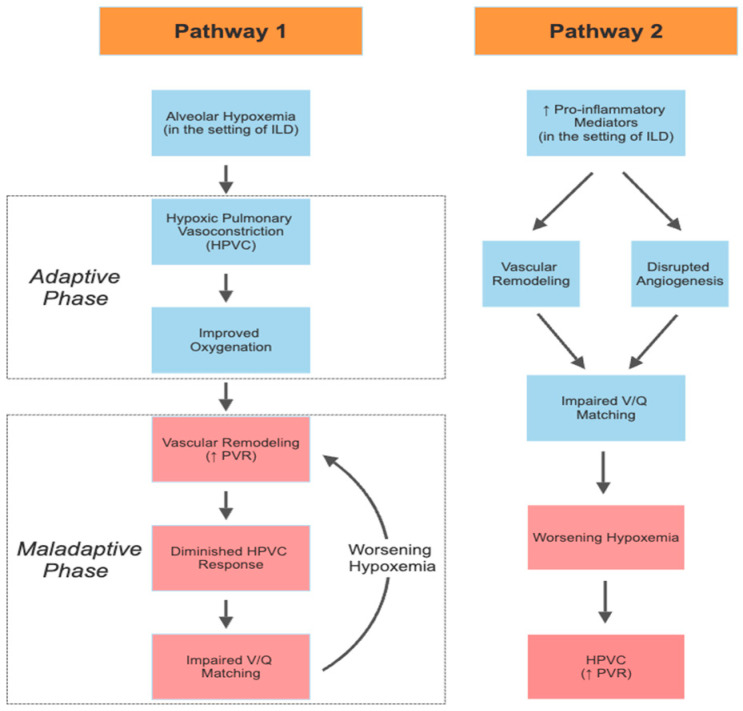
Proposed pathophysiologic pathways demonstrating progressive, worsening pulmonary hypertension in the setting of ILD. Pathway 1 demonstrates a proposed mechanism by which alveolar hypoxemia triggers hypoxic pulmonary vasoconstriction (HPVC). Initially adaptive, this mechanism improves oxygenation by diverting pulmonary blood flow to better ventilated regions. Over time, this process proceeds into a phase of maladaptive vascular remodeling, impairing HPVC and overall worsening ventilation/perfusion (V/Q) matching, progressing into a perpetual cycle of vascular remodeling and worsening oxygenation. Pathway 2 demonstrates the proposed pathway by which pro-inflammatory mediators in the setting of ILD promote vascular remodeling (independent of alveolar hypoxemia) and locally inhibit angiogenesis, ultimately progressing to worsening hypoxemia and elevations in PVR through HPVC.

**Figure 2 cells-12-02394-f002:**
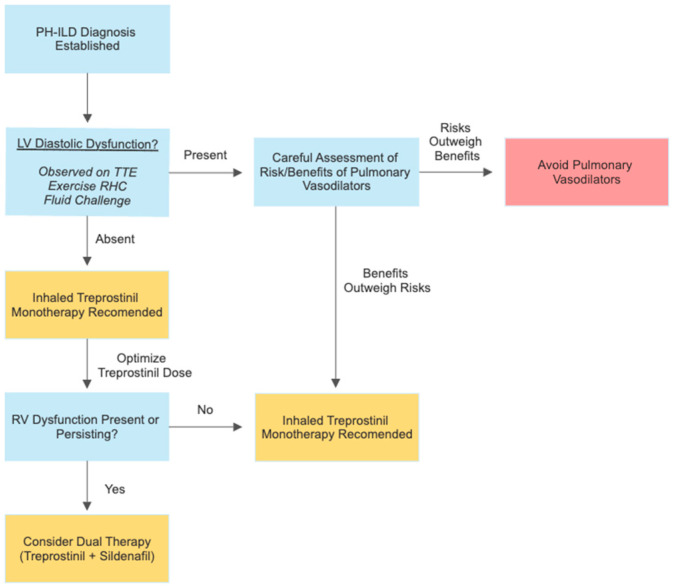
Considerations for the treatment of PH-ILD. After the careful assessment of a patient’s hemodynamic profile and diastolic function, inhaled treprostinil is recommended as a first line therapy for patients with PH-ILD based upon the results of the INCREASE trial. Treprostinil dosing should be optimized by titrating patients to the maximally tolerated dose (up to 64mcg inhaled four times per day in a dry powder inhaler). If right ventricular dysfunction is identified or persists after the optimization of a patient’s dose of inhaled treprostinil, clinicians should consider recommending dual therapy with inhaled treprostinil and sildenafil to patients. Patients with diastolic dysfunction may also benefit from inhaled treprostinil, though further research is necessary to establish safety and efficacy in this population. A careful discussion of the risks and benefits of therapy would need to be performed with patients before considering therapy in individuals with diastolic dysfunction. Legend: PH-ILD = pulmonary hypertension in interstitial lung disease, LV = left ventricular, TTE = transthoracic echocardiography, RV = right ventricular.

**Table 1 cells-12-02394-t001:** Findings Suggestive of Pulmonary Hypertension in Interstitial Lung Disease.

History	Syncope Dizziness Palpitations History of pulmonary embolism
Exam Findings	Jugular venous distension Peripheral edema Ascites Altered heart sounds (especially loud P2 or S2 heart sound) Hepatomegaly
Functional Testing	Severely reduced or worsening 6MWD Marked or worsening exertional desaturations Decreased heart rate recovery after exercise (<13 beats)
CT Imaging	RV:LV ratio >1 Increased PA:A ratio (>0.9)
Echocardiography	RV dilation Reduced TAPSE (<16 mm) RVOT diameter >3.4 cm Reduced RV fractional area change (<35%) Reduced RV ejection fraction on 3D echocardiography
PFTs	Severely reduced DLCO (<30%)
Laboratory Testing	Elevated BNP or NT-proBNP

Legend: 6MWD = six-minute walk distance, RV = right ventricle, LV = left ventricle, PA = pulmonary artery, A = aorta, TAPSE = tricuspid annular plane systolic excursion, RVOT = right ventricular outflow tract, 3D = three dimensional, BNP = brain naturetic peptide, NT-proBNP = N-terminal pro-brain naturetic peptide.

## Data Availability

Not applicable.

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
