# Peer review of "Pulmonary Hypertension in Interstitial Lung Disease: Updates in Disease, Diagnosis, and Therapeutics"

_cells, 2023, doi:10.3390/cells12192394_

Round 1

Reviewer 1 Report

The authors should be congratulated on the comprehensive, excellently written review on the very difficult subject concerning pulmonary hypertension in interstitial lung disease (PH-ILD). The epidemiology, pathophysiology, differential diagnosis and overview of the pharmacological trials performed in the population of PH-ILD is provided in a very condensed and well written manner. The document is enriched with good schemes and tables. The review is especially important and up to date in light of the positive results of the INCREASE trial providing data supporting the treatment of PH-ILD with inhaled Treprostinil. I have some minor remarks:

1)   In Table 1 in the echocardiography section provide the doppler measurements and the TAPSE / sPAP parameter and in the laboratory testing section the troponin level

2)   In Figure 2 the authors suggest that in a patient with concomitant PH-ILD and LV diastolic dysfunction inhaled treprostinil can be recommended if the benefits outweigh the risks – this concept should be elaborated

Author Response

Thank you for taking the time to review and provide feedback on our article on pulmonary hypertension in interstitial lung disease. We appreciate the insightful points which you have brought to our attention and hopefully we have addressed the highlighted issues adequately. Below you will find specific responses to the feedback you provided.

1)  Reviewer Feedback: “In Table 1 in the echocardiography section provide the doppler measurements and the TAPSE / sPAP parameter and in the laboratory testing section the troponin level”

Response: Thank you for these recommendations. We have added a comment to elaborate on an abnormal TAPSE value (<16mm). We have additionally elaborated on what defines a reduced fractional area change (<35%). In our view, monitoring troponin levels is not particularly useful when trying to identify cases of PH-ILD, as it is nonspecific markers of cardiac damage or dysfunction and can be generated through a multitude of mechanisms. Alternatively, elevations in BNP/NT-proBNP reflect increased RV strain (assuming normal LV function) and are more specific for presence of PH in patients with ILD. Though this association exists, “normal” cutoffs can vary between labs and with increasing age so we preferred using the term “elevated” in our table as opposed to a prespecified value.

2) Reviewer Feedback: “In Figure 2 the authors suggest that in a patient with concomitant PH-ILD and LV diastolic dysfunction inhaled treprostinil can be recommended if the benefits outweigh the risks – this concept should be elaborated”

Response: Thank you highlighting this gap in our manuscript. The decision to initiate therapy in this population can be highly challenging due to the lack of quality safety and efficacy data, as patients with LV dysfunction were excluded from the INCREASE trial. Despite this gap in our understanding, providers may feel compelled to consider therapy in patients with PH-ILD due to the limited targeted treatment options available with proven efficacy. We have added several lines of text (quoted below) discussing this gap in the literature and further elaborated in the caption for Figure 2.

“A careful consideration of the risks and benefits of inhaled treprostinil should also be given to patients with concurrent PH-ILD and left ventricular diastolic dysfunction before initiating therapy due to this population being excluded from the INCREASE trial. Initiation of pulmonary vasodilator therapy in these individuals can be inherently challenging, with their use potentially worsening left ventricular dysfunction due to increased left ventricular preload. Despite this concern, providers may feel inclined to consider therapy in this group of patients due to limited alternate treatment options. The exact characteristics of patients with favorable risk profiles remains unknown, though patients with mild diastolic dysfunction could conceivably be suited for therapy. Currently, therapeutic decisions must be individualized to each patients’ tolerable risk of decompensation and conceivable benefit from therapy in light of their PH-ILD severity. Close clinical monitoring will be necessary if therapy is offered.” (Page 10)

“...Patients with diastolic dysfunction may also benefit from inhaled treprostinil, though further research is necessary to establish safety and efficacy in this population. A careful discussion of the risks and benefits of therapy would need to be performed with patients before considering therapy in individuals with diastolic dysfunction.” (Figure 2 caption)

Reviewer 2 Report

In the present review, the authors intend to revise the current understanding of pathophysiology, updates in diagnostic approaches, and highlight recent clinical trials which provide a practical approach to medical management. However, the intended revision was presented rather superficially.

The main point is that the review lacks novelty. What is new or innovative in this publication about the many articles that deal with the same theme and have already been published (some of which very recently) in the literature?

The review falls short when, among other crucial topics, it fails to address in-depth issues like right ventricular dysfunction in PH-ILD or biomarkers of pulmonary hypertension in interstitial lung disease. The review also fails to discuss the use of phosphodiesterase type-5 inhibitors in PH-ILD adequately.

The authors must restructure the manuscript.

After careful consideration, it is not suitable for publication as it currently stands.

Extensive editing of the English language and style changes are required. 

Author Response

Thank you for taking the time to review and provide feedback on our article on pulmonary hypertension in interstitial lung disease. We appreciate the insightful points which you have brought to our attention and hopefully we have addressed the highlighted issues adequately. Below you will find specific responses to the feedback you provided.

1) Reviewer Feedback: “The main point is that the review lacks novelty. What is new or innovative in this publication about the many articles that deal with the same theme and have already been published (some of which very recently) in the literature?"

Response: There are a number of recent publications dealing with the same topic.   As the primary literature is limited, only so much novelty can be injected in this invited review.   We feel that our review provides a comprehensive review of diagnostic definitions and methods, theories in pathophysiology, and the history of negative trials preceding the recent approval of inhaled trepostinil for management. We additionally provide readers with recommendations for screening and pathways for management in light of the recent positive trial. We have also tried to highlight some of the gaps in knowledge in the literature, like the utility of PDE5i in PH-ILD and the treatment of mild PH-ILD that meets the current definition but not the inclusion/exclusion criteria of the INCREASE trial. 

2) Reviewer Feedback: "The review falls short when, among other crucial topics, it fails to address in-depth issues like right ventricular dysfunction in PH-ILD or biomarkers of pulmonary hypertension in interstitial lung disease."

Response: Thank you for highlighting this gap in our review. In regards to RV dysfunction, we do discuss the pathophysiology leading to RV failure in PH-ILD within our review on Page 4, Paragraph 2. In our initial version of the manuscript, we did not include a discussion of the management of RV dysfunction and chose to focus on management of patients’ pulmonary hypertension specifically given the complexities and breadth of this topic. In light of your feedback, we have expanded out discussion of RV dysfunction in PH (see quoted text below).

In terms of biomarkers, we have highlighted BNP/proBNP in Table 1 in our initial draft, though we recognize this was not discussed in the narrative text. Given this, we have highlighted some of the evidence of this biomarker in the section “Diagnosing a case of PH-ILD” (see quoted text below).

“In response to increasing PVR, patients with PH-ILD commonly demonstrate progressive right ventricular (RV) dysfunction. Relative to the left ventricle, the thin-walled RV is highly sensitive to hemodynamic changes and maintaining function is exquisitely important to support systemic cardiac output due to ventricular interdependence. With the progression of PH-ILD, RV afterload significantly increases due to increasing PVR, promoting progressive RV dysfunction. Chronically increased afterload leads to RV hypertrophy through neurohormonal activation with increased adrenergic tone.(21) This is initially adaptive, preserving stroke volume and increasing contractility, though over time beta adrenergic receptor density declines, adrenergic effectors are depleted, and tissues become less responsive to beta adrenergic agonists, forcing the RV to dilate to maintain stroke volume.(22) These processes ultimately lead to increasing RV wall stress, promoting RV ischemia and further declines in function. Continual stress, ischemia, and neurohormonal activation promote fibroblast collagen proliferation, resulting in progressive fibrosis, impaired diastolic relaxation, and diminished contractility.(21) Volume retention, worsening hypoxemia, and low cardiac output can result from this remodeling, placing patients with PH-ILD in the unfortunate scenario of right ventricular failure.

Although it is clear that advanced PH-ILD leads to RV dysfunction and this drives worsening outcomes, why relatively modest increases in pulmonary artery pressures (PAPs) that do not manifest in overt RV dysfunction are associated with such poor outcomes is less clear.  It is possible these patients rapidly progress to have more severe PH-ILD and studies fail to capture this progression over time.  Another possibility is that these patients have exercise-induced PH that causes RV strain with activity.  A final possibility in that these patients suffer from a relatively low cardiac output state that accounts for an elevated PVR despite relatively low mean PAPs.  These conditions could limit oxygen delivery to tissues with exertion.   Further study of patients with mild PH-ILD to determine the response to exercise and natural history is essential in answering these questions.  Furthermore, whether treatment with pulmonary vasodilators will halt the progression of mild PH-ILD and improve outcomes requires formal evaluation in clinical trials.”

“In regards to biomarkers of disease, brain natriuretic peptide (BNP) and N-terminal (NT) proBNP appear to have diagnostic utility. An NT-proBNP level of <95 ng/L has previously been suggested as a rule-out criteria for PH-ILD with a negative predictive value of 99%.(28) Alternatively, an elevated BNP or NT-proBNP value should trigger suspicion for underlying PH-ILD, but is nonspecific.(25)  It should be noted that obesity can lead to pseudo-normalization of BNP and pro-BNP values and can limit the utility of these assays as a screening tool.(29)

3) Reviewer Feedback: "The review also fails to discuss the use of phosphodiesterase type-5 inhibitors in PH-ILD adequately."

Response: Thank you for this feedback. On page 8 we reviewed pertinent trials and registries related to the use of PDE-5i in the management of PH-ILD. These included the STEP-IPF trial and the COMPERA registry. Both of these publications have flaws which limit their current application in the management of PH-ILD, notably a lack of clearly defined PH in the patient population or their retrospective nature. Given the lack of clearly positive prospective trials on the use of PDE-5 inhibitors, their use as second line or adjunctive therapy continues to be debatable. In regards to the limitations of the current literature, we have added text regarding potential evidence for the use of sildenafil in these patients and the need for further prospective trials on this topic. We have additionally restructured our discussion of PDE-5 inhibitors to again highlight the need for further investigation (see quoted text below).

“No prospective trial to date has demonstrated positive outcomes from the administration of phosphodiesterase-5 (PDE-5) inhibitors for the management of PH-ILD, though several trials and related registries suggest a potential benefit. Within the Sildenafil Trial of Exercise Performance in Idiopathic Pulmonary Fibrosis (STEP-IPF) trial, sildenafil failed to significantly improve 6MWD, though was found to improve numerous secondary outcomes, including oxygen saturation and DLCO.(50) Despite these findings, direct application of its results to patients with PH-ILD is difficult, as PH was not confirmed by RHC or TTE in this study and the presence of PH was simply enriched for through the use of DLCO cutoffs for enrollment. Alternatively, the COMPERA registry demonstrated improvements in 6MWD and functional class in patients with PH-ILD, of whom 88% received PDE-5 inhibitor therapy.(14) The subgroup analysis of patients with improved 6MWD from this registry also showed improved survival. Sildenafil has also demonstrated maintained V/Q matching in patients after administration, suggesting that it may preferentially dilate pulmonary vasculature in well ventilated regions.(40) Two recent studies compared combination of sildenafil with the anti-fibrotic medications pirfenidone and nintedanib versus anti-fibrotic therapy alone. Both studies failed to demonstrate a conclusive improvement of clinical outcomes from the use of sildenafil, although the INSTAGE study of nintedanib/sildenafil noted stabilization of BNP values in those treated with sildenafil.(51, 52) The authors of a recent meta-analysis including four studies of sildenafil in IPF suggests a potential mortality benefit of use of sildenafil in IPF; however, this conclusion should be viewed cautiously as the data failed to reach statistical significance.(53) Cumulatively, this data suggests that sildenafil is safe and may provide potential benefit in the management of PH-ILD, though further confirmatory data from prospective trials are needed.”

“Though monotherapy with inhaled treprostinil may be adequate to manage some individuals, patients with more advanced disease with severely increased PVR may require additional pharmacotherapy. Other patients may prove to be intolerant to inhaled treprostinil due to side effects, so alternative therapies may be considered. Recommendations for second line and adjunctive therapy are limited. The use of sildenafil appears to have the potential for benefit and safe use in some patients based upon the results of the COMPERA registry and subgroup analysis from the STEP-IPF trial.(14, 50) The authors believe cautious introduction of sildenafil in PH-ILD is reasonable to try in patients who are intolerant and unable to receive inhaled treprostinil, or as add-on therapy to prostanoids in severe PH-ILD (Figure 2).  Further prospective trials on this strategy are needed to better understand the of utility of sildenafil, and other PDE-5 inhibitors, in the management of severe PH-ILD.”

Round 2

Reviewer 2 Report

The authors attempted to respond adequately to the questions. Considering that most of the points raised in the review were satisfactorily addressed by the authors in the revised version, we believe the paper is now acceptable for publication in Cells.  

Minor English language and style changes are still required.